# Mechanized and Optimized Configuration Pattern of Crop-Mulberry Systems for Controlling Agricultural Non-Point Source Pollution on Sloping Farmland in the Three Gorges Reservoir Area, China

**DOI:** 10.3390/ijerph17103599

**Published:** 2020-05-20

**Authors:** Shouqin Zhong, Zhen Han, Jiangwen Li, Deti Xie, Qingyuan Yang, Jiupai Ni

**Affiliations:** 1School of Geographical Sciences, Southwest University, Chongqing 400715, China; zhong.qing.1988@163.com; 2State Cultivation Base of Eco-agriculture for Southwest Mountainous Land, Southwest University, Chongqing 400715, China; xdt@swu.edu.cn; 3College of resources and environment, Southwest University, Chongqing 400715, China; m13658372800@163.com (Z.H.); ljw0638@163.com (J.L.); 4Key Laboratory of Arable Land Conservation (Southwestern China), Ministry of Agriculture, Chongqing 400715, China

**Keywords:** agricultural non-point source pollution, sloping farmland, crop-mulberry system, optimized pattern

## Abstract

High-intensity utilization of sloping farmland causes serious soil erosion and agricultural non-point source pollution (AGNSP) in the Three Gorges Reservoir Area (TGRA). Crop-mulberry systems are important agroforestry systems for controlling soil, water, and nutrient losses. However, there are many different mulberry hedgerow planting patterns in the TGRA. In this study, soil structure, nutrient buildup, and runoff nutrient loss were observed in field runoff plots with five configurations: P1 (two longitudinal mulberry hedgerows), P2 (two mulberry contour hedgerows), P3 (three mulberry contour hedgerows), P4 (mulberry hedgerow border), and P5 (mulberry hedgerow border and one mulberry contour hedgerow), as well as a control (CT; no mulberry hedgerows). P1 had the smallest percentage of aggregate destruction (18.8%) and largest mean weight diameter (4.48 mm). P5 led to the greatest accumulation of ammonium nitrogen (NH_4_^+^–N) and total phosphorus (TP) (13.4 kg ha^−1^ and 1444.5 kg ha^−1^ on average, respectively), while P4 led to the greatest accumulation of available phosphorus (AP), nitrate nitrogen (NO_3_^−^–N), and total nitrogen (TN) (114.0, 14.9, and 1694.1 kg ha^−1^, respectively). P5 was best at preventing soil erosion, with the smallest average annual runoff and sediment loss of 112.2 m^3^ ha^−1^ and 0.06 t ha^−1^, respectively, which were over 72.4% and 87.4% lower than those in CT, respectively. P5 and P4 intercepted the most N in runoff, with average NH_4_^+^–N, NO_3_^−^–N, particulate N, and TN losses of approximately 0.09, 0.07, 0.41, and 0.58 kg ha^−1^, respectively, which were 49.7%, 76.2%, 71.3%, and 69.9% lower than those in CT, respectively. P5 intercepted the most P in runoff, with average TP and total dissolved phosphorus (TDP) losses of 0.09 and 0.04 kg ha^−1^, respectively, which were 77.5% and 70.4% lower than those in CT, respectively. Therefore, the pattern with one mulberry hedgerow border and one mulberry contour hedgerow (P5) best controlled AGNSP, followed by that with only a mulberry hedgerow border (P4).

## 1. Introduction

Water quality plays an important role in industry, agriculture, public health, and habitat protection [1]. Agricultural non-point source pollution (AGNSP) has been recognized as a key problem that significantly affects water quality worldwide, and efforts to control AGNSP from the field to watershed scale are ongoing [2,3]. To ensure the sustainability of agriculture and society, many studies have focused on AGNSP control and water quality improvement in river basins.

Soil, water, and nutrient losses on sloping farmland are the main sources of AGNSP, especially the losses of nitrogen (N) and phosphorus (P) [4,5,6,7]. At present, 50% of the world’s land area and 80% of its water bodies are affected by AGNSP, with approximately 75% of these areas impacted by N and P pollution [4]. The Three Gorges Dam (TGD), which is located on the Yangtze River, is the largest dam in the world. The ecological and environmental protection of the Three Gorges Reservoir Area (TGRA) plays a crucial role in the green development of the Yangtze River economic belt. However, the self-cleaning capacity of the Yangtze River has decreased, as the TGD has increased the water level and decreased water flow, further aggravating AGNSP and deteriorating water quality. Therefore, it is urgent to establish effective measures for controlling AGNSP, which seriously threatens the security of the ecological environment of the TGRA.

In recent years, contour hedgerows have been used as a major measure of soil, water, and nutrient conservation [8,9,10]. In the past 30 years, there have been many studies on contour hedgerows, all of which focused on contour hedgerow selection, soil erosion conservation, AGNSP control, ecological benefit evaluation, soil physical properties and nutrient changes, and other aspects. However, contour hedgerows are widely known for their ability to effectively control soil erosion and soil nutrient loss on sloping farmland. Thus, contour hedgerows are of major utility for controlling AGNSP.

Agroforestry is a sustainable agricultural system that involves the intentional combination of trees and shrubs with crops and livestock. As a kind of deciduous tree, mulberry, with its extremely developed root system, easy cultivation, and good ecological adaptability and resilience, has become one of the best choices for ecological restoration and environmental protection [9]. Farmers in the TGRA have followed a tradition of planting mulberry to cultivate silkworms since ancient times. Hence, mulberry has been widely planted in contour hedgerows, and has become a main economic crop in the TGRA [11]. The crop-mulberry system in the TGRA is a typical agroforestry system that can improve soil quality, control soil erosion, prevent land degradation, protect biodiversity, improve water quality, and regulate the climate [12,13,14,15,16]. At present, the pattern of planting crop-mulberry systems on sloping farmland has become one of the most promising farming patterns in the TGRA. However, there are many planting patterns for mulberry hedgerows, and the mechanisms and differences in the effects of these planting patterns on AGNSP control are unclear. Therefore, it is necessary to establish optimal planting patterns.

In this study on crop-mulberry systems, the changes in soil aggregate structure and nutrient buildup are quantified; the differences in the control of soil, water, nitrogen, and phosphorus losses between different crop-mulberry planting patterns on sloping farmland are comparatively analyzed; and optimal crop-mulberry planting patterns for controlling AGNSP on sloping farmland in the TGRA are proposed. The results provide a theoretical basis for popularizing the use of mulberry contour hedgerows to control soil and water losses, improve water quality, and maintain the sustainable development of the ecological environment in the TGRA.

## 2. Materials and Methods

### 2.1. Study Area

The TGRA (106°50′–110°07′E, 29°16′–31°41′N) is located in the belt of the eastern Sichuan Basin, has a total population of 30.75 million and covers a territory of 82,000 km^2^. The area has a predominantly low slope and hilly topography (Figure 1), with 95.7% of the total area consisting of mountains and hills at elevations of 175 m to approximately 2000 m. The central portion of the TGRA consists of parallel ridges and valleys. The southeastern and northeastern portions of this area are comprised of the Wuling Mountains and Qinba Mountains, respectively. More than 60% of the land in this area consists of sloping fields, and more than 70% of soils are purple soils (Cambisols or Regosols in the World Reference Base (WRB) Taxonomy) [17], with poor corrosion resistance. As the Yangtze River runs from southwest to northeast, the TGRA is home to 18 secondary tributaries, including tributaries of the Jialing River, Wujiang River, Daning River, and Xiangxi River, etc. The climate is classified as a subtropical humid monsoon climate, with an annual average temperature of 17–20 °C and an annual average rainfall of 1000–1200 mm. The rainfall is concentrated in summer, mostly in the form of rainstorms. The unique ecological environmental conditions in this reservoir area are very conducive to the occurrence of AGNSP.

### 2.2. Experimental Design

The experiment was conducted at the TGRA AGNSP monitoring station in Fuling district, Chongqing municipality (Figure 1). Eighteen test plots (width × length = 5.0 × 10.0 m) were used for observations of soil, runoff, and sediment. According to the main mulberry hedgerow planting patterns currently used in the TGRA, five crop-mulberry configuration patterns—namely, pattern 1 (two longitudinal mulberry hedgerows; P1), pattern 2 (two mulberry contour hedgerows; P2), pattern 3 (three mulberry contour hedgerows; P3), pattern 4 (mulberry hedgerow border; P4), pattern 5 (mulberry hedgerow border and one mulberry contour hedgerow; P5), and a control (no mulberry hedgerows; CT) were established (Figure 2). Three replicates were designed for each treatment. The belt width of each mulberry hedgerow was 0.5 m. The planting mode was a mustard–corn rotation. The amount of fertilizer applied to each plot was consistent, and N, P, and K were applied at approximately 156.8, 89.9, and 152.7 kg ha^−1^, respectively, for mustard planting, and 143.9, 41.0, and 108.7 kg ha^−1^, respectively, for corn planting.

### 2.3. Sampling and Testing

After the mustard tuber harvest (February 2013 and February 2014) and corn harvest (July 2013 and August 2014), soil samples were collected from 0–20 cm and 20–40 cm at different sampling positions (Figure 2) in the plots, and a 2.0 kg mixed soil sample was obtained from three replicates at each sampling point. After the stones, plant roots, animal residues, and gravel were removed, the soil samples were air-dried at room temperature, ground, mixed, sieved through a 2 mm nylon sieve, and stored in plastic bags. At the same time, a ring knife (volume of 200 cm^3^) was used to collect undisturbed soil samples that were used to measure the soil bulk density.

The rock fragment content was measured by washing with water [18], the soil particle size distribution (PSD) and microaggregate composition were determined by the pipette method, according to Stokes’ law [19], and the abundance of water-stable aggregates was measured according to the dry sieving and wet sieving method [20]. Soil total nitrogen (TN) was measured by the Kjeldahl method [21], and available nitrogen (AN) was determined by alkaline hydrolysis diffusion [22]. Total phosphorous (TP) was analyzed by dissolving soil samples in NaOH solution, followed by molybdenum–stibium colorimetry [23]. Available phosphorous (AP) was determined by using 0.50 mol/L sodium bicarbonate (NaHCO_3_) extraction–molybdenum-antimony anti-colorimetry [24]. For the samples that were passed through the 1 mm mesh, soil ammonium nitrogen (NH_4_^+^–N) and nitrate nitrogen (NO_3_^−^–N) were measured using colorimetry and analyzed by automatic flow injection [23]. All experiments were repeated at least three times, and the results are expressed as the mean values.

Runoff and sediment were collected after each natural rainfall event. First, the runoff and sediment were fully mixed, and three 500 mL slurry samples were collected with a stratified sampler from each plot. Then, the three slurry samples were mixed together and divided into two subsamples: one was used for sediment determination, with oven drying at 105 °C, and the other was centrifuged and filtered to analyze the TN, TP, NH_4_^+^–N, and NO_3_^−^–N in the runoff. The TN and TP in runoff were measured with alkaline potassium persulfate oxidation, and the total dissolved nitrogen (TDN), total dissolved phosphorus (TDP), NH_4_^+^–N, and NO_3_^−^–N in the runoff were determined using a flow injection analyzer, with the samples filtered by a 0.45 μm filter membrane. Particulate nitrogen (PN) was obtained by subtracting the TDN from the TN.

### 2.4. Data Analysis

The mean weight diameter (MWD) of water-stable soil aggregates and the percentage of aggregate destruction (PAD) of soil aggregates with a particle diameter >0.25 mm were calculated according to [25]. Soil nutrient buildup and runoff nutrient loss were estimated with the method proposed in [11]. Significant differences between the soil properties, runoff, sediment, and nutrient losses among the treatments were statistically analyzed by multiple analysis of variance (ANOVA) *(p* < 0.05) to test the effects of slope position.

## 3. Results

### 3.1. Effects of Crop-Mulberry Systems on Soil Aggregate Structure

Different treatments had different influences on the soil aggregate structure in the crop-mulberry systems. The abundances of water-stable aggregates larger than 0.25 mm in P1–P5 were 1.26, 1.04, 1.11, 1.10, and 1.09 times that in CT, respectively. P1 had the greatest abundance of water-stable aggregates larger than 0.25 mm (806.33 g kg^−1^), and the water-stable aggregates larger than 0.5 mm accounted for 76.20% of the total water-stable aggregates in this treatment (Figure 3a). The PAD values in P1–P5 were 54.38%, 91.75%, 82.72%, 79.39%, and 85.11% of the PAD value in CT, respectively, with P1 exhibiting the smallest PAD (18.82%). At the same time, the MWD of water-stable aggregates was largest in P1 (4.48 mm) (Figure 3b).

Different treatments also had different effects on the soil microaggregate composition (Figure 3c). The percentages of 0.25–2.00 mm aggregates in P1 and P4 were significantly higher than that in CT. The percentage of 0.05–0.25 mm microaggregates in CT was the largest (18.11%). There were no significant differences in the percentages of 0.02–0.05 mm microaggregates between the treatments. The abundance of 0.002–0.020 mm microaggregates in P1 were significantly greater than in P5. The abundances of <0.002 mm microaggregates in P4 and P5 was significantly greater than that in P2. In P1, the most abundant soil aggregate sizes were 0.25–2.00 mm and 0.002–0.020 mm, with values of 7.95% and 45.12%, respectively; these were approximately 4.36%–58.35% and 10.84%–22.70% higher than those in the other treatments. The percentage of <0.002 mm microaggregates was the highest in P4 (17.54%); specifically, it was 14.20%, 31.47%, 24.91%, 1.48%, and 37.74% higher than in P1, P2, P3, P5, and CT, respectively.

The soil particle composition differed among the different crop-mulberry configuration patterns (Figure 3d). The abundance of 0.25–2.00 mm soil particles in P4 was significantly greater than in P1, P3, P5, and CT. The abundance of 0.05–0.25 mm soil particles in P5 was significantly greater than in P1. There were no significant differences in the abundances of 0.02–0.05 mm soil particles between treatments. The abundances of 0.002–0.020 mm soil particles in P2 and CT were significantly greater than those in P1, P3, P4, and P5. The abundance of clay particles (<0.002 mm) in P1 was the highest (35.55%); specifically, it was 31.56%, 18.09%, 16.01%, 9.11%, and 36.34% greater than that in P2–P5 and CT, respectively. Therefore, mulberry hedgerows effectively prevented the loss of clay particles in soil and improved the structure of soil aggregates. Among the treatments, P1 improved the soil aggregate structure the most.

### 3.2. Effects of Crop-Mulberry Systems on Soil Nutrient Buildup

The crop-mulberry systems improved soil nutrients, but the effects of different treatments on soil N and P were inconsistent. During the two years of the test period, the soil AN buildup in P1–P5 was between 82.9 and 121.0 kg ha^−1^, which was approximately 36.4%, −6.5%, 13.3%, 30.4%, and 27.3% higher than in CT (88.7 kg ha^−1^), respectively (Figure 4a). The soil AP buildup in P1–P5 was between 56.8 and 114.0 kg ha^−1^, which was approximately 28.8%, −34.3%, 7.9%, 31.9%, and 31.0% higher than in CT (86.4 kg ha^−1^), respectively (Figure 4b). The soil NH_4_^+^–N buildup in P1–P5 was between 6.0 and 13.4 kg ha^−1^, which was approximately 9.3%, −46.2%, −18.4%, 2.2%, and 19.5% higher than that in CT (11.2 kg ha^−1^), respectively (Figure 4c). The soil NO_3_^−^–N buildup in P1–P5 was between 7.9 and 14.9 kg ha^−1^, which was approximately −0.9%, −19.0%, 29.4%, 52.7%, and 40.6% higher than that in CT (9.8 kg ha^−1^), respectively (Figure 4d). The soil TN buildup in P1–P5 was between 1484.2 and 1694.0 kg ha^−1^, which was approximately 24.6%, 29.0%, 13.2%, 29.2%, and 20.9% higher than that in CT (1311.0 kg ha^−1^), respectively (Figure 4e). The soil TP buildup in P1–P5 was between 1276.3 and 1444.5 kg ha^−1^, which was approximately 34.6%, 30.1%, 23.6%, 20.7%, and 36.6% higher than that in CT (1057.4 kg ha^−1^), respectively (Figure 4f). Therefore, compared with CT, P1, P3, P4, and P5 effectively improved soil N and P buildup, including the buildup of AN, AP, NH_4_^+^–N, NO_3_^−^–N, TN, and TP. P2 effectively increased the buildup of only soil TN and TP. At the same time, P1 increased soil AN buildup the most; P4 increased soil NO_3_^−^–N, TN, and AP buildup the most; and P5 increased soil NH_4_^+^–N and TP buildup the most.

### 3.3. Effects of Crop-Mulberry Systems on Water and Soil Losses

The runoff and sediment in each treatment in 2014 were similar to those in 2013 (Figure 5), and there were significant differences in runoff and sediment between treatments (*p* < 0.05). In 2013, the runoff in P1–P5 was between 113.1 and 342.1 m^3^ ha^−1^, which was approximately 68.1%, 15.4%, 35.6%, 71.8%, and 72.0% higher than that in CT (404.6 m^3^ ha^−1^), respectively; the sediment in P1–P5 was between 0.08 and 0.23 t ha^−1^, approximately 70.5%, 47.7%, 68.2%, 75.0%, and 81.8% higher than that in CT (0.44 t ha^−1^), respectively. In 2014, the runoff in P1–P5 was between 111.2 and 324.3 m^3^ ha^−1^, approximately 68.7%, 21.0%, 49.3%, 64.0%, and 72.9% higher than that in CT (410.6 m^3^ ha^−1^), respectively; the sediment in P1–P5 was between 0.04 and 0.18 t ha^−1^, which was approximately 87.5%, 67.9%, 80.4%, 87.5%, and 92.9% higher than that in CT (0.56 t ha^−1^), respectively. Therefore, P1–P5 intercepted surface runoff and sediment, which increased with mulberry growth (age). Moreover, the interception of sediment was greater than that of runoff, and P5 intercepted the most surface runoff and sediment. The relationship between runoff and sediment was linear (*R*^2^ = 0.796; Table 1).

### 3.4. Effects of Crop-Mulberry Systems on Runoff Nutrient Loss

There were large differences in runoff N loss among the crop-mulberry planting patterns (Figure 6). Significance analysis revealed significant differences in the TN and PN losses in 2013 and TN loss in 2014, but there were no significant differences in NH_4_^+^–N or NO_3_^−^–N loss. The runoff TN losses in CT, P2, and P3 in 2013 were significantly greater than those in P1, P4, and P5, and the distribution of PN was similar to that of TN. The runoff TN losses in CT and P2 in 2014 were significantly greater than that in P1. The losses of NH_4_^+^–N in P1–P5 were between 0.02 and 0.25 kg ha^−1^, with an average value of 0.07–0.13 kg ha^−1^. Compared with the average loss of NH_4_^+^–N in CT (0.18 kg ha^−1^), the average losses of NH_4_^+^–N in P1–P5 were reduced by 23.7%, 58.3%, 49.1%, 49.4%, and 50.0%, respectively. The losses of NO_3_^−^–N in P1–P5 were between 0.02 and 0.28 kg ha^−1^, with an average value of 0.07–0.19 kg ha^−1^. Compared with the average loss of NO_3_^−^–N in CT (0.29 kg ha^−1^), the average losses of NO_3_^−^–N in P1–P5 were reduced by 66.2%, 36.2%, 58.6%, 76.6%, and 75.9%, respectively. The losses of PN in P1–P5 were between 0.34 and 2.25 kg ha^−1^, with an average value of 0.4–1.56 kg ha^−1^. Compared with the average loss of PN in CT (1.45 kg ha^−1^), the average losses of PN in P1–P5 were reduced by 72.5%, –7.7%, 22.6%, 71.7%, and 71.0%, respectively. The losses of TN in P1–P5 were between 0.38 and 2.72 kg ha^−1^, with an average value of 0.58–1.84 kg ha^−1^. Compared with the average loss of TN in CT (1.94 kg ha^−1^), the average losses of TN in P1–P5 were reduced by 67.2%, 5.2%, 30.4%, 70.2%, and 69.5%, respectively. Linear fitting revealed the linear relationships of NO_3_^−^–N, TN, and PN with runoff, with *R*^2^ values of 0.923, 0.965, and 0.906, respectively (Table 1). In general, the P4 and P5 treatments intercepted the most N in runoff.

The five different crop-mulberry planting patterns intercepted varying degrees of P in runoff (Figure 7). Compared with those in 2013, the losses of TDP and TP in runoff in 2014 decreased by 41.5% and 20.1% on average, respectively. The runoff TP and TDP losses in CT and P1 were significantly higher than those in P5. The average losses of TP in P1–P5 were between 0.09 and 0.22 kg ha^−1^. Compared with the average loss of TP in CT (0.40 kg ha^−1^), the average losses of TP in P1–P5 were reduced by 45.6%, 50.3%, 59.1%, 58.8%, and 77.5%, respectively. The average losses of TDP in P1–P5 were between 0.04 and 0.08 kg ha^−1^. Compared with the average loss of TDP in CT (0.14 kg ha^−1^), the average losses of TDP in P1–P5 were reduced by 46.3%, 40.7%, 52.0%, 50.7%, and 70.4%, respectively. Therefore, P5 had the best interception effect on P loss via runoff.

## 4. Discussion

### 4.1. Crop-Mulberry Systems Effectively Improve Soil Internal Structure and Increase Soil Nutrient Buildup

As the basic unit of soil structure, soil aggregates play an important role in regulating the physical and chemical properties, fertility, and ecological functions of soil. Soil aggregate structure and stability are closely related to many ecological and environmental problems [26,27,28]. Good soil aggregate structure has a positive effect on supporting animal and plant life, regulating soil moisture, slowing greenhouse gas emissions, preventing soil erosion, and controlling AGNSP [26,29,30]. In this study, P1–P5, with different mulberry planting modes, effectively improved the structure of soil aggregates, and P1 improved the soil aggregate structure the most. After hedgerow planting, organic plant residues are returned to the soil, and plant roots and soil animals increase, which promotes an increase in organic cementing substances in soil, thereby increasing the stability of soil aggregates and conserving soil and water [31,32,33,34,35]. The amount of space for mulberry root growth in P1 was greater than that in the other planting patterns, so the root systems in P1 were the most developed. Therefore, this treatment improved the soil aggregate structure the most. However, since the mulberry hedgerows in P1 were planted along a slope, the interception of runoff, sediment, and nutrients in this treatment was reduced to some extent.

The main sources of AGNSP are the loss of P and P nutrients in farmland soil [4,36,37]. When the amount of fertilizer applied to soil is fixed, increasing the accumulation of soil nutrients can reduce the loss of soil nutrients with runoff [9,11]. Crop-mulberry systems can improve soil nutrient buildup, whereas there have been significant differences in this effect among the different planting modes [8,11,38]. Compared with CT, P1, P3, P4, and P5 effectively improved soil N and P buildup, such as the soil buildup of AN, AP, NH_4_^+^–N, NO_3_^−^–N, TN, and TP. The P2 treatment effectively increased the buildup of only soil TN and TP. The P5 configuration mode increased the buildup of NH_4_^+^–N and TP in soil the most, while P4 improved the buildup of soil AP, NO_3_^−^–N and TN the most. Thus, P5 and P4 are recommended to prevent nutrient loss and increase soil nutrient buildup in farmland with poor soil fertility and steep slopes.

### 4.2. Crop-Mulberry Systems Effectively Control the External Loss of Soil, Water, and Nutrients

Contour hedgerows are an important measure used to maintain water and soil, reduce N and P losses, and control AGNSP, and are widely used in mountainous and hilly areas [8,9,39,40,41]. The establishment of contour hedgerows in different places may have different effects on controlling the loss of soil, water, and nutrients [41]. On arable sloping land, alfalfa contour hedgerows and toon contour hedgerows have been found to effectively reduce runoff by 34% and 26%, sediment by 86% and 84%, N losses by 50% and 42%, and P losses and 68% and 64% [42]. Likewise, Wang et al. [8] found that with contour hedgerows of alfalfa and *Hemerocallis citrina* Baroni, TN loss decreased by 81% and 85%, and TP loss decreased by 91% and 93%, respectively. Thus, contour hedgerows provide more options for controlling AGNSP based on the local resources in hilly and mountainous areas.

In the TGRA, residents of rural areas plant mulberry and raise silkworms. Hence, as one of the main economic crops, mulberry can be easily found and used in contour hedgerows to reduce soil, water, and nutrient losses on sloping farmland in the TGRA [9,11]. This study employed mulberry in contour hedgerows, based on traditional farming habits and the ecological concept of adjusting measures on the basis of local conditions. In this study, the control effects of mulberry hedgerows with different configurations on runoff on sloping farmland were studied. The results showed that compared with CT, P1–P5 decreased runoff by 68.4%, 18.3%, 42.5%, 67.9%, and 72.5%, respectively, and sediment by approximately 80.0%, 59.0%, 75.0%, 82.0%, and 88.0%, respectively. Hence, mulberry hedges have a greater effect on sediment interception than on runoff interception, which is consistent with the findings of previous studies on contour hedgerows [11,42]. Furthermore, there were large differences in runoff N and P losses among the crop-mulberry planting patterns. Compared with CT, P1–P5 reduced NH_4_^+^–N, NO_3_^−^–N, PN, TN, TP, and TDP by an average of 46.1%, 62.7%, 46.0%, 48.5%, 58.3%, and 52.0%, respectively. Overall, P5 intercepted the most soil, water, P, and N on sloping farmland in the TGRA, followed by P4. Thus, other conditions (such as topography) that permit the use of the P5 pattern (mulberry hedgerow border and one mulberry contour hedgerow) are recommended to reduce AGNSP on sloping farmland in the TGRA.

## 5. Conclusions

In this study, the effects of five crop-mulberry configuration patterns (P1–P5, compared with CT) on controlling AGNSP on sloping farmland were analyzed. Specifically, soil aggregate structure, soil nutrient buildup, water and soil losses, and nutrient loss via runoff were observed in field runoff plots. Overall, compared with CT, the five different configuration patterns (P1–P5) improved internal features (soil structure and nutrient buildup) and effectively controlled soil, water, and nutrient losses, thereby reducing the occurrence of AGNSP. The P5 configuration mode had the best effect on increasing the buildup of NH_4_^+^–N and TP in the soil and controlling soil, water, TP, and TDP losses; the P4 configuration mode had the best effect on improving the buildup of soil AP, NO_3_^−^–N, and TN, as well as on controlling the loss of nitrogen (NH_4_^+^–N, NO_3_^−^–N, TN, and PN) in runoff; and the P1 configuration mode had the best effect on improving soil aggregate structure. In addition, mulberry hedgerows had a greater effect on the interception of sediment than on the interception of runoff and its nutrients. Therefore, the configuration pattern consisting of a mulberry hedgerow border and one mulberry contour hedgerow (P5) is the best configuration pattern for controlling AGNSP on sloping farmland in the TGRA, followed by the configuration pattern consisting of a mulberry hedgerow border (P4). At the same time, planting herbaceous plants, such as alfalfa and vetiver grass, under hedges of mulberry plants is suggested to further strengthen the interception of slope runoff and its nutrients.

## Figures and Tables

**Figure 1 ijerph-17-03599-f001:**
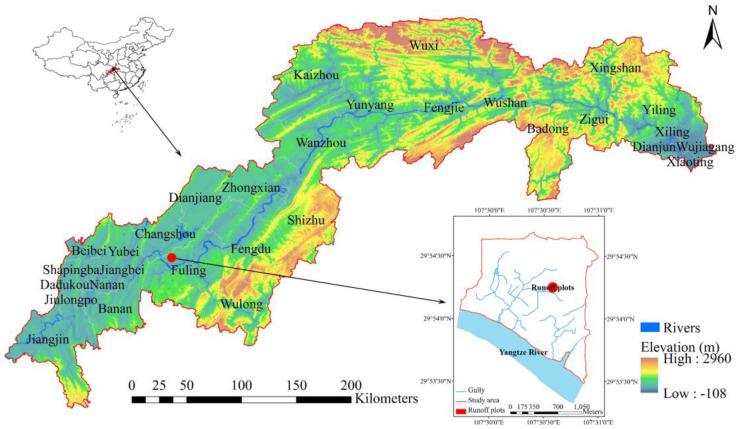
Location of the study area and experimental site.

**Figure 2 ijerph-17-03599-f002:**
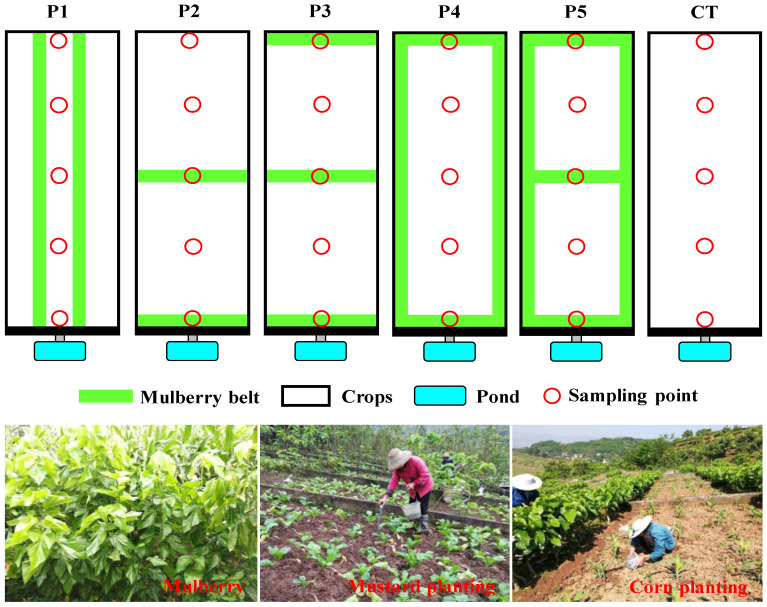
Schematic diagram of different crop-mulberry configuration patterns. P1: two longitudinal mulberry hedgerows; P2: two mulberry contour hedgerows; P3: three mulberry contour hedgerows; P4: mulberry hedgerow border; P5: mulberry hedgerow border and one mulberry contour hedgerow; CT: no mulberry hedgerows, as a control. The width of each mulberry belt was 0.5 m.

**Figure 3 ijerph-17-03599-f003:**
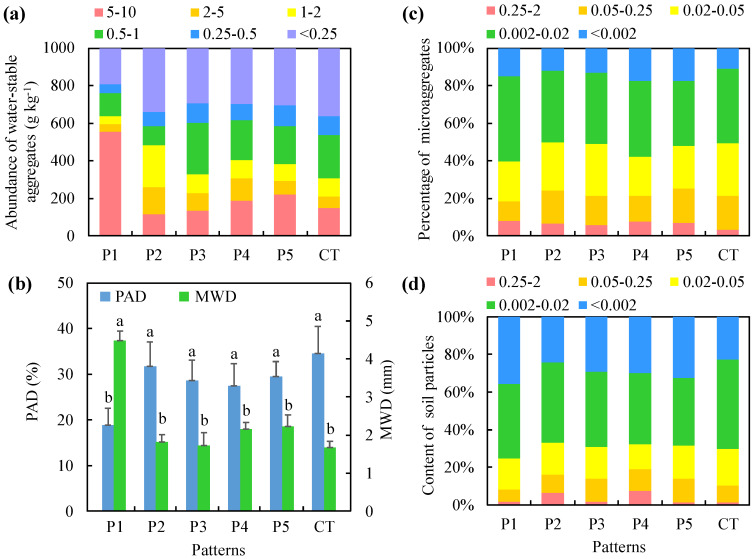
Soil aggregate structure and particle characteristics under different crop-mulberry planting patterns. (**a**) Water-stable aggregates; (**b**) mean weight diameter (MWD) of water-stable aggregates and percentage of aggregate destruction (PAD) for soil aggregates with a particle diameter >0.25 mm; (**c**) microaggregates; (**d**) soil particle composition. The same letters above the bars indicate no significant difference and different letters indicate significant differences between these groups *(p* < 0.05).

**Figure 4 ijerph-17-03599-f004:**
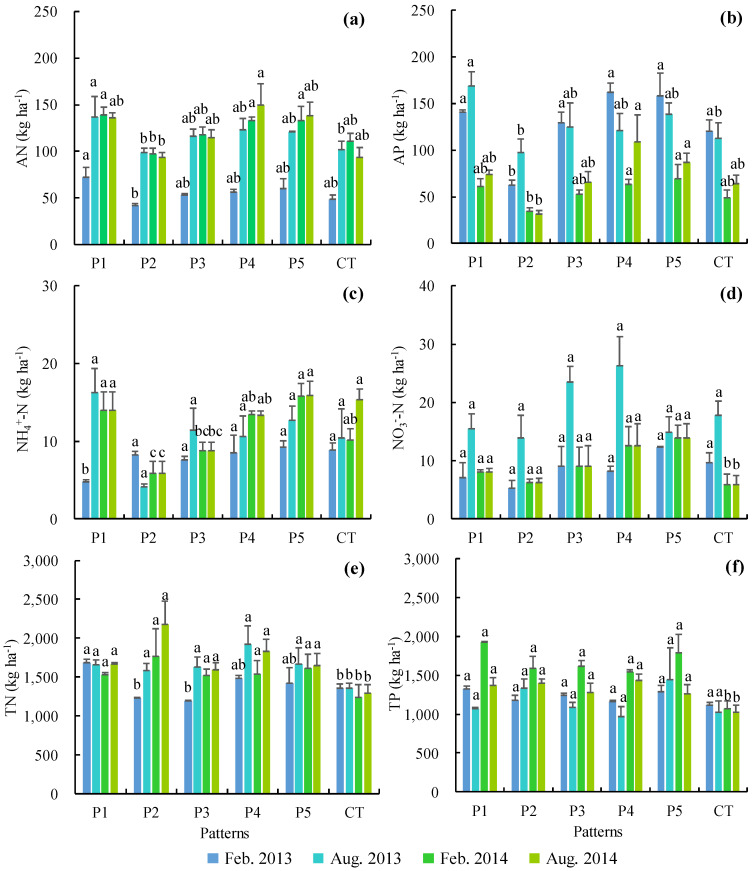
Soil buildup of available nitrogen (AN) (**a**), available phosphorous (AP) (**b**), NH_4_^+^–N (**c**), NO_3_^−^–N (**d**), total nitrogen (TN) (**e**), and total phosphorous (TP) (**f**) in the crop-mulberry systems on sloping farmland. The same letters above the bars indicate no significant difference and different letters indicate significant differences between these groups *(p* < 0.05).

**Figure 5 ijerph-17-03599-f005:**
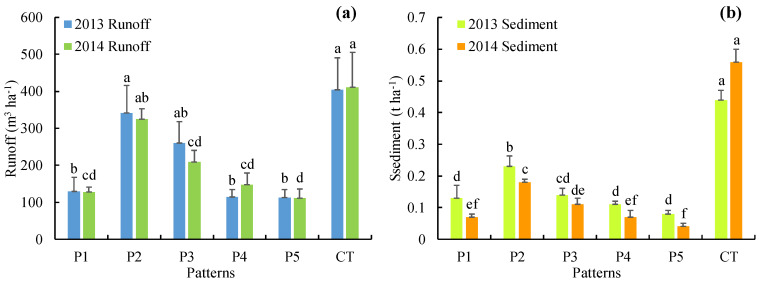
Runoff and sediment for different crop-mulberry planting patterns. The same letters above the bars indicate no significant difference and different letters indicate significant differences between these groups *(p* < 0.05).

**Figure 6 ijerph-17-03599-f006:**
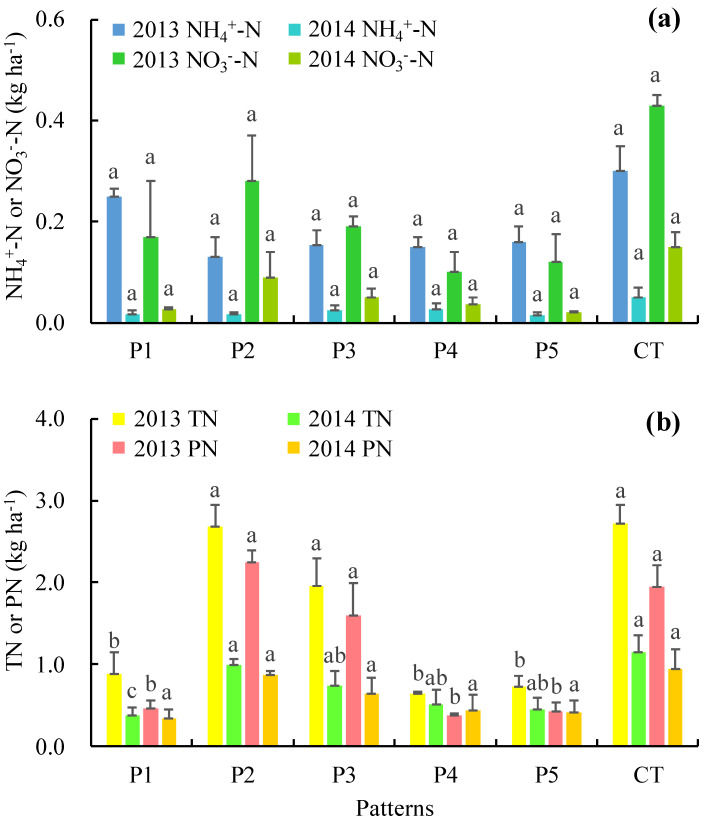
Nitrogen (N) loss from runoff under different crop-mulberry planting patterns. The same letters above the bars indicate no significant difference and different letters indicate significant differences between these groups *(p* < 0.05).

**Figure 7 ijerph-17-03599-f007:**
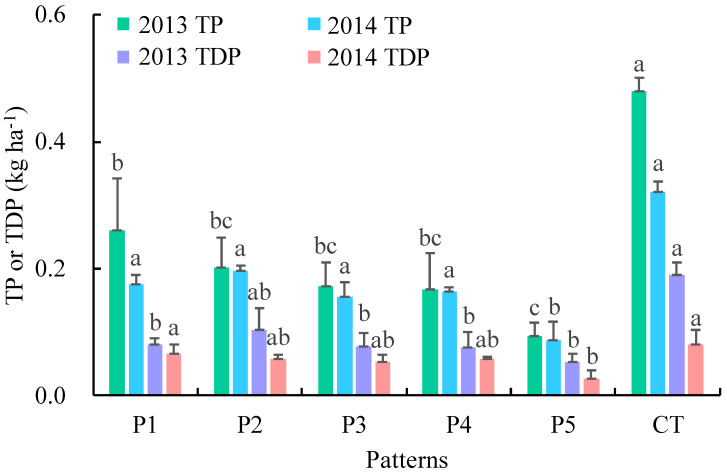
Phosphorus (P) loss via runoff under different crop-mulberry planting patterns. The same letters above the bars indicate no significant difference and different letters indicate significant differences between these groups *(p* < 0.05).

**Table 1 ijerph-17-03599-t001:** Linear fitting of runoff with the losses of sediment, NH_4_^+^–N, NO_3_^−^–N, TN, PN, TP, and TDP.

Variable	Equation	*R*²
Sediment	*y*_1_ = 0.0012*r* − 0.0869	0.7957
NH_4_^+^–N	*y*_2_ = 0.0001*r* + 0.0783	0.1735
NO_3_^−^–N	*y*_3_ = 0.0007*r* − 0.0116	0.9232
TN	*y*_4_ = 0.0051*r* + 0.0124	0.9651
PN	*y*_5_ = 0.0042*r* − 0.0592	0.9062
TP	*y*_6_ = 0.0007*r* + 0.0578	0.6031
TDP	*y*_7_ = 0.0002*r* + 0.028	0.7045

Note: *r* represents runoff (m^3^ ha^−1^); *y*_1–_*y*_7_ represent the losses of sediment (t ha^−1^), NH_4_^+^–N, NO_3_^−^–N, TN, PN, TP, and TDP (kg ha^−1^), respectively, via runoff; *R* represents the linear correlation coefficient.

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
