# Peer review of "Mechanized and Optimized Configuration Pattern of Crop-Mulberry Systems for Controlling Agricultural Non-Point Source Pollution on Sloping Farmland in the Three Gorges Reservoir Area, China"

_ijerph, 2020, doi:10.3390/ijerph17103599_

Round 1

Reviewer 1 Report

Thank you for giving me the chance to review this paper.

Nutrient and soil losses are important problems in the cropland.

The Ms designed experiments in a crop-mulberry system (i.e., 5 major crop-mulberry configuration patterns were established). Changes in soil aggregate structure and nutrient buildup are quantified, the differences in controlling soil, water, nitrogen and phosphorus losses between different crop-mulberry planting patterns on sloping farmland are comparatively analyzed, and optimal crop-mulberry planting patterns for controlling AGNSP on sloping farmland in the TGRA are finally proposed.

The Ms is overall interesting and writes well. While the experimental design is not clear, revisions need to be addressed.

Runoff and sediment were collected after each rainfall event. However, did you simulate the rainfall or is it natural rainfall? When or how much the rainfall is? It is not clear, while it is very important for your study.

Details:

1.What is the main question addressed by the research? Is it relevant and interesting?

The Ms studies the soil, runoff and nutrients under five major major crop-mulberry configuration patterns in a crop-mulberry system, and optimal crop-mulberry planting patterns for controlling AGNSP on sloping farmland in the TGRA are proposed. This would be interesting for cropland in a slope area.

2.How original is the topic? What does it add to the subject area compared with other published material?

The study comparatively explores five major crop-mulberry configuration patterns in a crop-mulberry system of the Three Gorges Reservoir region on soil, runoff and nutrients. The study would be closely relevant to AGNSP controlling on sloping farmland, and provide a theoretical basis for popularizing the use of mulberry contour hedgerows for soil and water loss controls.

3.Is the paper well written? Is the text clear and easy to read?

English needs to be improved.

  1. Are the conclusions consistent with the evidence and arguments presented? Do they address the main question posed?

Yes

Author Response

Reviewer 1

(1) General comments:

Thank you for giving me the chance to review this paper.

Nutrient and soil losses are important problems in the cropland.

The Ms designed experiments in a crop-mulberry system (i.e., 5 major crop-mulberry configuration patterns were established). Changes in soil aggregate structure and nutrient buildup are quantified, the differences in controlling soil, water, nitrogen and phosphorus losses between different crop-mulberry planting patterns on sloping farmland are comparatively analyzed, and optimal crop-mulberry planting patterns for controlling AGNSP on sloping farmland in the TGRA are finally proposed.

The Ms is overall interesting and writes well. While the experimental design is not clear, revisions need to be addressed.

Runoff and sediment were collected after each rainfall event. However, did you simulate the rainfall or is it natural rainfall? When or how much the rainfall is? It is not clear, while it is very important for your study.

Response: Thank you for your positive comments and valuable suggestions regarding our manuscript. We have revised the text according to your comments. Runoff and sediment were collected after each natural rainfall event. First, the runoff and sediment were fully mixed, and then three 500-mL slurry samples were collected with a stratified sampler from each plot. Please see lines 150-152.

(2) Details:

Comment 1: What is the main question addressed by the research? Is it relevant and interesting?

The Ms studies the soil, runoff and nutrients under five major major crop-mulberry configuration patterns in a crop-mulberry system, and optimal crop-mulberry planting patterns for controlling AGNSP on sloping farmland in the TGRA are proposed. This would be interesting for cropland in a slope area.

Response: Thank you very much for your comments.

Comment 2: How original is the topic? What does it add to the subject area compared with other published material?

The study comparatively explores five major crop-mulberry configuration patterns in a crop-mulberry system of the Three Gorges Reservoir region on soil, runoff and nutrients. The study would be closely relevant to AGNSP controlling on sloping farmland, and provide a theoretical basis for popularizing the use of mulberry contour hedgerows for soil and water loss controls.

Response: Thank you for your comments.

Comment 3: Is the paper well written? Is the text clear and easy to read?

English needs to be improved.

Response: According to your comment, the English writing of the manuscript has been carefully edited by native English-speaking editors at American Journal Experts.

Comment 4: Are the conclusions consistent with the evidence and arguments presented? Do they address the main question posed? Yes

Response: Thank you very much for your positive comments.

Reviewer 2 Report

Your findings were instructive – and supportive of the works of others (Wang, Zhu, and Xia).  However, the way in which you presented the data makes it difficult to comprehend the points you want to make.

The main issue is how you present the data.  There is too much data in the paragraphs, making them hard-to-understand.  It might be better to place the data in tables and then only discuss the relevant, meaningful, or significant findings in the text.  If you want to keep the data in the text, maybe use smaller paragraphs with only one or two sets of data and more explanation to differentiate the different statistics.

There is also a problem with the abstract – it has too much information. It is over 800 words, three to four times as large as an abstract should be. The abstract should be a few highlights and key insights.  But what you have included is really a summary that should be placed at the end of the research findings.

When you do make revisions, pay attention to the how you use adverbs to introduce sentences. In the current paper, you use “therefore” in successive sentences (Line 91 and 93).  You also use “However” when it appears you are in agreement with another research finding (Line 337) – the better word choice where would be “Likewise.”

Finally, it appears one of the ranges of microaggregates was misstated – “0.02-0.02 mm” (Line 193).

Once this paper as been recast to improve its presentation (and readability), it should be ready to be accepted for publication.

Author Response

Reviewer 2

Comments and Suggestions for Authors:

Comment 1: Your findings were instructive – and supportive of the works of others (Wang, Zhu, and Xia). However, the way in which you presented the data makes it difficult to comprehend the points you want to make.

Response: Thank you for your comments regarding our manuscript. We have carefully revised the manuscript according to your comments.

Comment 2: The main issue is how you present the data. There is too much data in the paragraphs, making them hard-to-understand. It might be better to place the data in tables and then only discuss the relevant, meaningful, or significant findings in the text. If you want to keep the data in the text, maybe use smaller paragraphs with only one or two sets of data and more explanation to differentiate the different statistics.

Response: Thank you very much. According to your comments, we have revised the results section. The data analyzed in the results are mainly displayed in the form of a figure, which more intuitively to represents the patterns among the data. In addition, we have deleted the detailed data and mainly discussed the meaningful and significant findings in the results.

Comment 3: There is also a problem with the abstract – it has too much information. It is over 800 words, three to four times as large as an abstract should be. The abstract should be a few highlights and key insights. But what you have included is really a summary that should be placed at the end of the research findings.

Response: Thank you for your comment. According to your comment, we have revised the abstract. Please see lines 17-39.

Comment 4: When you do make revisions, pay attention to the how you use adverbs to introduce sentences. In the current paper, you use “therefore” in successive sentences (Line 91 and 93). You also use “However” when it appears you are in agreement with another research finding (Line 337) – the better word choice where would be “Likewise.”

Response: Thank you. According to your comment, we have revised the adverbs in the text; please see lines 82 and 324.

Comment 5: Finally, it appears one of the ranges of microaggregates was misstated – “0.02-0.02 mm” (Line 193).

Response: Thank you very much. We have revised the text according to your comment. Please see line 182.

Comment 6: Once this paper as been recast to improve its presentation (and readability), it should be ready to be accepted for publication.

Response: Thank you very much for your positive comments and valuable suggestions regarding our manuscript.

Round 2

Reviewer 2 Report

The paper is a now a much better read. The changes and clarifications allow the important findings to shine through. 

It is always heartening as a reviewer to see a paper with interesting and important findings reach its potential and become publishable.

In my opinion, “Mechanism and optimized configuration pattern of crop-mulberry systems for controlling agricultural non-point source pollution on sloping farmland in the Three Gorges Reservoir Area, China” has accomplished that – and the authors should be commended for making the improvements to the current draft of the paper (compared to the original submission).